# HYDROPT: An Open-Source Framework for Fast Inverse Modelling of Multi- and Hyperspectral Observations from Oceans, Coastal and Inland Waters

**Tadzio Holtrop** [1,2,*] and **Hendrik Jan Van Der Woerd** [1]

1 Department of Water & Climate Risk, Institute for Environmental Studies (IVM), Vrije Universiteit, De Boelelaan 1111, 1081 HV Amsterdam, The Netherlands; h.j.vander.woerd@vu.nl
2 Department of Freshwater and Marine Ecology (FAME), Institute for Biodiversity and Ecosystem Dynamics, University of Amsterdam, P.O. Box 94240, 1090 GE Amsterdam, The Netherlands
* Correspondence: tadzio.holtrop@icloud.com

**Abstract:** Biomass estimation of multiple phytoplankton groups from remote sensing reflectance spectra requires inversion models that go beyond the traditional band-ratio techniques. To achieve this objective retrieval models are needed that are rooted in radiative transfer (RT) theory and exploit the full spectral information for the inversion. HydroLight numerical solutions of the radiative transfer equation are well suited to support this inversion. We present a fast and flexible Python framework for forward and inverse modelling of multi- and hyperspectral observations, by further extending the formerly developed HydroLight Optimization (HYDROPT) algorithm. Computation time of the inversion is greatly reduced using polynomial interpolation of the radiative transfer solutions, while at the same time maintaining high accuracy. Additional features of HYDROPT are specification of sensor viewing geometries, solar zenith angle and multiple optical components with distinct inherent optical properties (IOP). Uncertainty estimates and goodness-of-fit metrics are simultaneously derived for the inversion routines. The pursuit to retrieve multiple phytoplankton groups from remotely sensed observations illustrates the need for such flexible retrieval algorithms that allow for the configuration of IOP models characteristic for the region of interest. The updated HYDROPT framework allows for more than three components to be fitted, such as multiple phytoplankton types with distinct absorption and backscatter characteristics. We showcase our model by evaluating the performance of retrievals from simulated $R_{rs}$ spectra to obtain estimates of 3 phytoplankton size classes in addition to CDOM and detrital matter. Moreover, we demonstrate HYDROPTs capability for the inter-comparison of retrievals using different sensor band settings including coupling to full spectral coverage, as would be needed for NASA's PACE mission. The HYDROPT framework is now made available as an open-source Python package.

**Keywords:** HYDROPT; ocean color; radiative transfer; hyperspectral; inversion; phytoplankton size class; NASA PACE

## 1. Introduction

Ocean-color remote sensing has opened up the opportunity to monitor the biological and chemical processes of the ocean on an unprecedented scale. Satellite sensors continue to provide a synoptic view of ocean biogeochemistry at high spatial and temporal resolution that would be impossible to acquire through in-situ sampling campaigns [1].

More than four decades ago the first ocean-color instrument, the Coastal Zone Color Scanner (CZCS; a list of abbreviations is given under section "Abbreviations") was launched and for the first time provided a detailed picture of phytoplankton dynamics in the upper layers of the ocean. The main product derived from CZCS observations were estimates of the primary pigment found in almost all phytoplankton, chlorophyll-a [1]. The first order variation in remote sensing reflectance ($R_{rs}$) in open ocean waters is due to chlorophyll-a [2]

characterized by two absorption peaks in the blue and in the red, causing a greening of the waters with increasing phytoplankton concentrations.

However, chlorophyll-a alone does not give the full picture of phytoplankton diversity [3]. The diversity in phytoplankton is characterized by different physiological and morphological traits that affect biogeochemical processes and the ecological niches that these species inhabit [4,5]. In turn, these physiological and morphological differences between phytoplankton groups affect the optical properties of the water column and in theory could be detected in the remotely sensed signal. Exploiting this second order variation in the remotely sensed signal to derive a more detailed description of phytoplankton community structure has become a top priority in the ocean-color community [3,6].

Many physiological traits of phytoplankton, which in turn affect ecological and biogeochemical processes, are correlated with cell size and pigment composition [4,5,7]. The smallest phytoplankton such as *Prochlorococcus* and *Synechococcus* are numerically the most abundant phytoplankton in the ocean and are important contributors to global primary production [8]. The largest phytoplankton such as diatoms dominate in cold, nutrient rich waters and are important contributors to the biological pump through efficient draw down of carbon to the sea floor [9]. In addition to chlorophyll-a, *Prochlorococcus* contains (divinyl) chlorophyll-b, *Synechococcus* contains a diversity of phycobilin pigments, and diatoms contain chlorophyll-c and fucoxanthin, with which they exploit different parts of the light spectrum [7,10].

Cell size and pigment composition affect absorption and backscatter characteristics of phytoplankton [9–12]. The packaging effect is an important driver of the variability in phytoplankton absorption which is depended on phytoplankton cell size and intracellular pigment concentration [10]. Numerous studies have investigated the link between community size structure and chlorophyll-a specific absorption: the phytoplankton absorption normalized to chlorophyll-a concentration derived from HPLC-based measurements. The chlorophyll-a specific absorption, especially around the blue absorption peak (440–490 nm), decreases with increasing cell size (Figure 1) [11,13]. Small-celled phytoplankton communities are characterized by pronounced absorption peaks whereas communities dominated by larger cells exhibit less pronounced peaks. Brewin et al. [14] empirically derived backscatter coefficients for phytoplankton communities of different dominating cell sizes. Phytoplankton communities dominated by the largest cells show a low and spectrally flat backscatter, whereas smaller celled communities exhibit elevated backscatter at shorter wavelengths and exponentially decreasing scatter with increasing wavelength (Figure 2). The differences in absorption and backscatter characteristics could in theory be exploited to detect and discriminate the size structure of phytoplankton groups from the remotely sensed signal [6].

However, efforts to exploit the full spectral reflectance to obtain phytoplankton community structure have been challenging and have led to the identification of certain phytoplankton groups without estimates of biomass [2,3,15,16]. Most of the semi-analytical retrieval models only decompose either the retrieved absorption or backscatter coefficients to obtain information on community structure and ignore the inter-dependency between the IOPs and measured $R_{rs}$ signal. The empirical models suffer from noise in the detected pigment concentrations of representative species and are biased to the data collected. Inversion methods based on radiative transfer principles are needed to accurately separate the contribution of various phytoplankton groups to $R_{rs}$ [15,17]. HydroLight [18] is the state-of-the-art radiative transfer model for ocean-color applications; however, it is computational costly and can therefore not be used to invert $R_{rs}$ spectra in near real time.

The HYDROPT model [19] was developed to overcome the time intensive computations of RT simulations and at the same time sustain highly accurate $R_{rs}$ calculations over several wavelengths. HYDROPT is based on HydroLight RT simulations and speed and accuracy are realized through interpolation of the radiative transfer solutions. The inversion is achieved by finding the best fit to the measured $R_{rs}$ by means of nonlinear optimization techniques. The original HYDROPT algorithm was adapted to coastal and

inland waters and was only capable of fitting three optical components to a limited set of 9 wavebands from the Medium Resolution Imaging Spectrometer (MERIS). Coastal and inland waters are characterized by high levels of absorption and scattering by CDOM and particulate matter, leaving only one component to represent the absorption and scattering budget of the phytoplankton population. To retrieve multiple phytoplankton groups, HYDROPT needs to be updated to accommodate several optical spectral models for different phytoplankton groups in addition to CDOM and detrital matter. The increasing number of ocean-observing sensors also necessitate the need for flexible waveband configuration beyond MERIS band settings. A final impetus to this research has been the publication of new absorption values of pure water in the 250–550 nm wavelength range [20], a part of the spectrum important for the discrimination of phytoplankton groups [4,7].

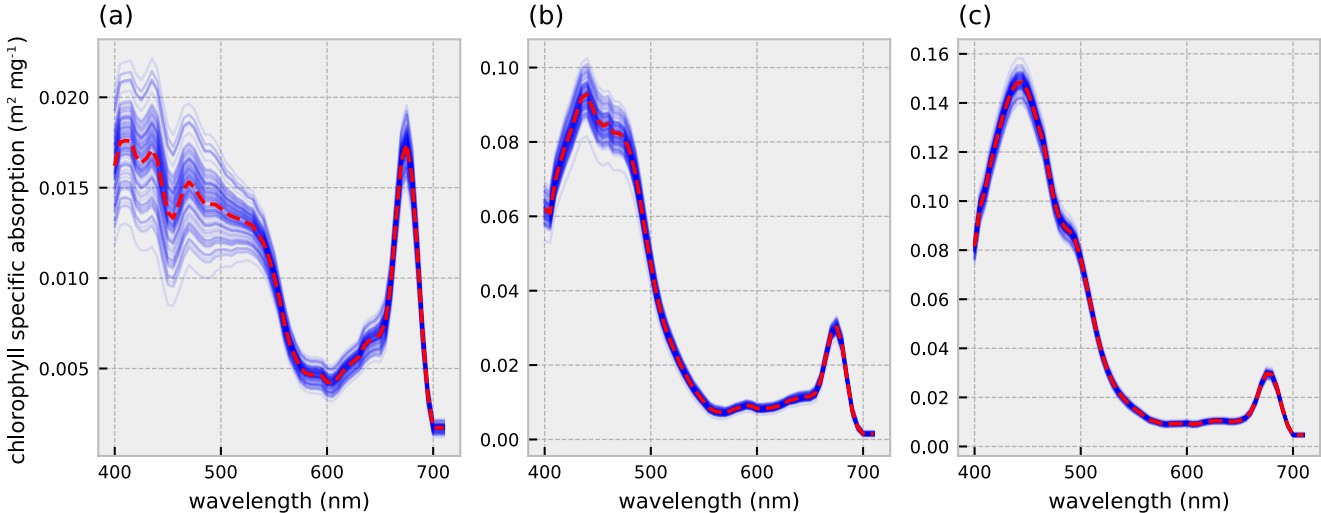

**Figure 1.** Chlorophyll-a specific absorption for (**a**) micro- (**b**) nano- and (**c**) pico-phytoplankton. A hundred absorption spectra are randomly sampled to visualize variability within and across size classes (blue lines). Mean absorption spectra are used as for the inversion with HYDROPT (red dashed lines). Absorption coefficients and standard errors are obtained from Uitz et al. [11].

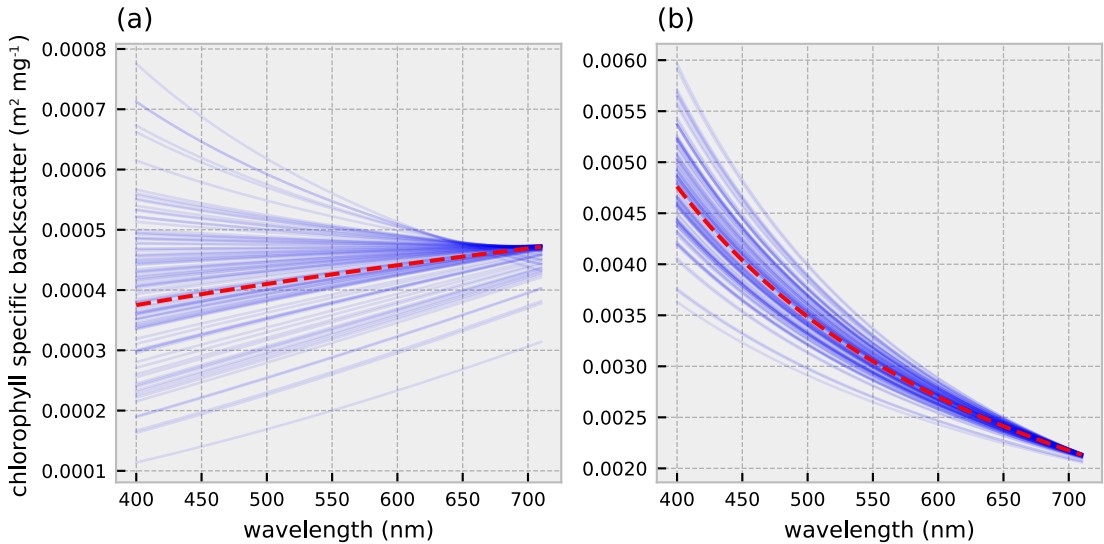

**Figure 2.** Chlorophyll-a specific backscatter for (**a**) micro- (**b**) nano- and pico-phytoplankton. A hundred backscatter spectra are randomly sampled to visualize variability within and across size classes (blue lines). Mean backscatter spectra are used for the inversion with HYDROPT (red dashed lines). Backscatter coefficients and standard errors are obtained from Table 3-database D in Brewin et al. [12].

The objective of this research is three-fold. First, we present the update to the formerly developed HYDROPT algorithm [19]. HYDROPT has been completely rewritten in Python and is made open-source [21]. The most notable update is the coupling to full spectral coverage as would be needed for hyperspectral missions like NASA's Plankton, Aerosol, Cloud, ocean Ecosystem (PACE) mission [22]. HYDROPT is sensor agnostic, allowing flexible band settings in the 400–710 nm range while accounting for the bidirectional nature of $R_{rs}$ [23–25] . Second, we test the feasibility of the framework to retrieve three phytoplankton size classes in addition to CDOM and detrital matter from simulated $R_{rs}$ spectra. The dataset incorporates changes in phytoplankton community structure in addition to variable mass specific IOPs for phytoplankton, detrital matter and CDOM to best represent the optical variation encountered in natural water bodies [5,11,26,27]. Third, we demonstrate the applicability of HYDROPTs flexible waveband settings to both multi- and hyperspectral sensors and we compare the performance of retrievals for Sea-viewing Wide Field-of-view Sensor (SeaWiFS), Ocean and Land Color Instrument (OLCI) and the PACE ocean-color sensor.

## 2. Materials and Methods

### 2.1. HydroLight Simulations

HydroLight forward simulation are run to obtain the remote sensing reflectance, $R_{rs}$ (units: $sr^{-1}$), as a function of IOPs, viewing geometry ($\theta_v$, $\phi_v$) and solar zenith angle ($\theta_s$):

$$R_{rs}(\lambda, \theta_v, \phi_v, \theta_s) = \frac{L_w(\lambda, \theta_v, \phi_v, \theta_s)}{E_d(\lambda)} \qquad (1)$$

where $\lambda$ is the wavelength, $L_w$ (units: $Wm^{-2}sr^{-1}nm^{-1}$) the water-leaving radiance and $E_d$ (units: $Wm^{-2}nm^{-1}$) the downwelling irradiance just above the water surface. The remote sensing reflectance is calculated for a viewing geometry consistent with the standard HydroLight quad layout with a 10° resolution in nadir viewing angle ($\theta_v$) and 15° resolution in azimuthal angle ($\phi_v$). The RT simulations are run at a 5 nm resolution between 400 and 710 nm and for solar zenith angles between 0–80° at a 10° resolution.

For the forward simulations a bio-optical model is chosen that includes the IOP of water, CDOM, mineral particles and phytoplankton [18]. The goal of the forward radiative transfer simulations is to obtain $R_{rs}$ for a realistic set of absorption and backscatter coefficients. Details on the IOP spectral models, wind and sky conditions can be found in the supplementary information (Table A1). Fluorescence and Raman scattering are ignored. A lookup table (LUT) is constructed that contains all permutations of CDOM absorption at 440 nm ($\rho_{CDOM}$: 0.005–10 $m^{-1}$) and concentrations of minerals ($\rho_{min}$: 0.01–100 g $m^{-3}$) and phytoplankton ($\rho_{chl}$: 0.01–32 mg $m^{-3}$). Every component is varied over 10 logarithmic spaced intervals between their respective minimum and maximum values, yielding a total of a 1000 simulations per waveband for every viewing geometry.

### 2.2. Polynomial Forward Model

HYDROPT uses a polynomial function to describe the relationship between $R_{rs}$ and the IOPs of the water column, the total absorption ($a$) and backscatter ($b_b$) coefficients (units: $m^{-1}$). Since the last version of the framework [19], the use of the total scattering coefficient has been replaced by the backscatter coefficient. Through polynomial interpolation of the RT solutions a fast and accurate forward model is constructed that takes into account the bidirectional nature of $R_{rs}$. The rationale behind a polynomial description of the RT solution space is that it provides an analytical form of the relationship between $R_{rs}$ and IOPs, the total spectral absorption and backscatter [19]. Since polynomial expressions are easily differentiable, they provide a fast way to invert reflectance spectra in conjunction with gradient-based optimization routines. The following polynomial form is chosen to relate $R_{rs}$ to the IOPs:

$$R'_{rs}(\lambda) \approx \sum_{i=0}^{n} \sum_{j=0}^{n} c_{ij}(\lambda) \, (a')^i \, (b'_b)^j \qquad (2)$$

where $R'_{rs}$ is the natural log-transformed reflectance at wavelength $\lambda$, $n$ is the degree of the polynomial expression, $c_{ij}(\lambda)$ is the fitted coefficient of the corresponding polynomial term and $a'$ and $b'_b$ are the log-transformed absorption and backscatter coefficients. The powers of the polynomial terms are indicated by $i$ and $j$. The (log-transformed) reflectance values, $R'_{rs}$, together with the absorption and backscatter coefficients, $a'$ and $b'_b$, are obtained from HydroLight simulations. It should be noted that $R_{rs}$ and the coefficients $c_{ij}$ are also dependent on the viewing geometry and sky conditions but are omitted from Equation (2) for brevity. The polynomial coefficients $c$ are linearly interpolated to obtain $R_{rs}$ for any waveband, and viewing geometry.

Using Equation (2) the remote sensing reflectance can now be calculated for any combination of water constituents given that:

$$
\begin{aligned}
a &= a_w + \sum_{i=1}^{m} a_i^* \, \rho_i \\
b_b &= b_{b,w} + \sum_{i=1}^{m} b_{b,i}^* \, \rho_i
\end{aligned}
\tag{3}
$$

where $a$ and $b_b$ are the total absorption and backscatter coefficients respectively, $a_w$ and $b_{b,w}$ are the absorption and backscatter due to water and $a_i^*$ and $b_{b,i}^*$ are the mass specific absorption and backscatter coefficients for constituent $i$ and $\rho$ is the concentration of the constituent. Multiple optical constituents ($m$) can be used to calculate $R_{rs}$ such as different phytoplankton groups as long as the total absorption and backscatter values are within the same range as the IOP values used in the radiative transfer simulations.

Given the matrix $C$ of all coefficients $c_{ij}$ and the matrix $X$ of the transformed polynomial terms of $a'$ and $b'_b$, Equation (2) can be rewritten in vector notation:

$$
R'_{rs} \approx \mathrm{diag}(C \cdot X)
\tag{4}
$$

To prevent over-fitting, the degree of the polynomial model, $n$, is determined through 10-fold cross-validation. Since reflectance values differ in order of magnitude between wavebands, the root mean squared relative error ($RMSRE$) is used as a measure of accuracy [28,29]:

$$
RMSRE = \sqrt{\frac{1}{k} \sum_{\lambda=400}^{710} \left[ \frac{R_{rs}(\lambda) - \hat{R_{rs}}(\lambda)}{R_{rs}(\lambda)} \right]^2}
\tag{5}
$$

where $k$ is the number of wavebands ($k = 63$), $R_{rs}$ the reflectance given by HydroLight and $\hat{R_{rs}}$ the reflectance predicted by our forward model (Equations (2) and (4)). The root mean squared relative error scales the errors across all wavebands. The 1-SE rule is used to select the most parsimonious model within one standard error of the best model [30].

### 2.3. Optimization

The estimation of IOPs and concentration of aquatic optical constituents from $R_{rs}$ relies on minimizing the difference between the observed ($R_{rs}$) and predicted ($\hat{R_{rs}}$) reflectance spectra. The loss function to be minimized is therefore expressed as:

$$
\chi^2 = \sum_{i=1}^{k} \left[ \frac{R_{rs}(\lambda_i) - \hat{R_{rs}}(\lambda_i, \rho_m)}{\sigma_i} \right]^2
\tag{6}
$$

where $\rho_m$ is the absorption or concentration of water constituent $m$ (e.g., CDOM, non-algal particles, chlorophyll), $k$ the number of wavebands and $\sigma_i$ the weight assigned to waveband $i$. The concentration of constituents that minimize the loss function are assumed to be the best estimate for that respective reflectance spectrum. Gradient-based optimization routines, such as Levenberg-Marquardt, use the gradient of the loss function to converge to a local or global solution. Several Python packages implement these optimization

routines [31,32] and either approximate the gradient numerically or allow the user to supply an analytical expression of the gradient. Since numerical approximations are computationally more costly and time consuming, providing an analytical expression of the gradient is preferred. Here we derive the gradient of $R_{rs}$ w.r.t the constituent concentrations $\rho$:

$$\nabla R_{rs}(\lambda_i, \rho_m) = \left[ \frac{\partial R_{rs}(\lambda_i)}{\partial \rho_1}, \cdots, \frac{\partial R_{rs}(\lambda_i)}{\partial \rho_m} \right] \tag{7}$$

where $\nabla R_{rs}$ is the gradient for waveband $i$ w.r.t constituents 1 to $m$ and $\partial R_{rs}/\partial \rho_m$ is the partial derivative of $R_{rs}$ w.r.t constituent $m$. The partial derivatives can be further decomposed as follows:

$$\frac{\partial R_{rs}}{\partial \rho_m} = \frac{\partial R_{rs}}{\partial R'_{rs}} \left( \frac{\partial R'_{rs}}{\partial a'} \frac{\partial a'}{\partial a} \frac{\partial a}{\rho_m} + \frac{\partial R'_{rs}}{\partial b'_b} \frac{\partial b'_b}{\partial b_b} \frac{\partial b_b}{\rho_m} \right) \tag{8}$$

with

$$\frac{\partial R_{rs}}{\partial R'_{rs}} = \frac{\partial R_{rs}}{\partial ln(R_{rs})} = R_{rs} \tag{9}$$

and $\partial R'_{rs}/\partial a'$ and $\partial R'_{rs}/\partial b'_b$ are the first order derivative of the forward model (Equation (2)) given by:

$$\frac{\partial R'_{rs}}{\partial a'} = \sum_{j=0}^{n} \sum_{i=0}^{n} i \, c_{ij} (a')^{i-1} (b'_b)^j$$

$$\frac{\partial R'_{rs}}{\partial b'_b} = \sum_{j=0}^{n} \sum_{i=0}^{n} j \, c_{ij} (b'_b)^{j-1} (a')^i \tag{10}$$

The partial derivatives $\partial a'/\partial a$ and $\partial b'/\partial b$ are simply:

$$\frac{\partial a'}{\partial a} = \frac{\partial ln(a)}{\partial a} = \frac{1}{a}$$

$$\frac{\partial b'_b}{\partial b_b} = \frac{\partial ln(b_b)}{\partial b_b} = \frac{1}{b_b} \tag{11}$$

The two terms $\partial a/\partial \rho_m$ and $\partial b_b/\partial \rho_m$ denote the first order derivative of the absorption and backscatter model for constituent $m$. In the case of a linear model as in Equation (3) the derivatives reduce to $a_m^*$ and $b_{b,m}^*$ respectively.

Evaluating the gradient in Equation (7) for every waveband $\lambda$ and constituent $\rho$ gives the Jacobian of our forward model to be used in gradient-based optimization routines:

$$J = \begin{bmatrix} \nabla R_{rs}(\lambda_1, \rho_m) \\ \vdots \\ \nabla R_{rs}(\lambda_n, \rho_m) \end{bmatrix} \tag{12}$$

The Levenberg-Marquardt implementation also estimates the co-variance matrix for every retrieval [32]. Given the co-variance matrix $C$, the relative retrieval error for constituent $i$ can be approximated as follows:

$$\delta_i \approx \frac{\sqrt{C_{i,i}}}{\hat{\rho}_i} \tag{13}$$

where $C_{i,i}$ is the variance in the estimation of constituent $i$ and $\hat{\rho}_i$ is the estimated concentration of constituent $i$.

### 2.4. IOCCG Dataset

To test the generalizability of the polynomial forward model we first validate the model against an independent dataset that assumes different optical conditions under

which the simulations have been conducted. Here we assess HYDROPT's capability to accurately predict $R_{rs}$ with changes in the particle volume scattering function (VSF). The VSF, and thus the backscatter ratio, is extremely difficult to determine in the field and laboratory. Most studies have relied on approximations of the VSF by either interpolating the VSF between angles that can be measured in the field [33] or through theoretical calculations [34]. Because of the uncertainty in the VSF of marine particles, several assumptions are made when performing in-water radiative transfer calculations. For the HydroLight simulations in this study we assumed a Fournier–Forand (FF) phase function with a 1.4% backscatter ratio for phytoplankton as well as for minerals. The well-established IOCCG dataset [27] comprises forward simulations with different phase functions for phytoplankton and particulate matter: a FF phase function with a 1% backscatter ratio for phytoplankon and a Petzold phase function with a 1.8% backscatter ratio for minerals. To establish generalizability or our forward model, the absorption and backscatter coefficients from the IOCCG simulations are used as input to the polynomial model to predict $R_{rs}$. The predicted $R_{rs}$ spectra by HYDROPT are validated against the IOCCG $R_{rs}$ values.

### 2.5. Hyperspectral Phytoplankton Size Class Dataset

To establish the potential of HYDROPT to retrieve multiple phytoplankton size classes from $R_{rs}$, we create a hyperspectral dataset containing simulated spectra [35]. The dataset takes into account the natural variability in chlorophyll-a and mass specific IOPs for three phytoplankton size classes as well as detritus and CDOM absorption. In addition, the dataset also incorporates the co-variation between the optical components as would be encountered in the case-I waters [27].

By combining HPLC samples from open ocean waters and diagnostic pigment analysis, Uitz et al. [11] estimated chlorophyll-a specific absorption spectra for three different phytoplankton size classes: pico- (<2 μm), nano- (2–20 μm) and micro-phytoplankton (>20 μm) (Figure 1). One hundred spectra for each size class are sampled from the distribution of absorption coefficients in Uitz et al. [11] (Figure 1). These spectra are randomly selected and used for forward modelling of our dataset. For the inversion part of the exercise, we only supply the average chlorophyll-a specific absorption spectra (red dashed lines in Figure 1) to HYDROPT.

Brewin et al. [12] estimated the chlorophyll-a specific backscatter coefficients for these three phytoplankton size classes using several in-situ databases that contain HPLC measurements and particulate backscatter coefficients. By applying diagnostic pigment analysis to the HPLC measurements, biomass for each size class could be estimated and subsequently their fractional contribution to the total particulate backscatter [12]. The mass specific backscatter is described by a power-law according to:

$$b_{bp}^*(\lambda) = b_{bp}^*(\lambda_0)\left(\frac{\lambda}{\lambda_0}\right)^{\gamma_1} \tag{14}$$

where $b_{bp}^*$ (units: m$^2$ mg$^{-1}$) is the mass specific backscatter coefficient, $\lambda_0$ is the reference wavelength at 470 nm and $\gamma_1$ is the spectral slope. The mass specific backscatter coefficient, spectral slope and variability in the estimates of these coefficients were determined by non-linear least squares fitting and bootstrapping. Here, we model phytoplankton backscatter using the parameters described in Table 3-database D in [12]. The given 95% confidence intervals are converted to standard errors assuming normally distributed errors, yielding the hundred spectral backscatter curves in Figure 2. Estimates of spectral backscatter of pico- and nano-phytoplankton did not yield significantly different spectral backscatter curves and are therefore grouped together (Figure 2b) [12].

Several studies have derived empirical relationships between the phytoplankton community size structure and total chlorophyll-a [36,37]. These population models assume that small-celled phytoplankton dominate at low chlorophyll concentrations and large cells at high chlorophyll concentrations [38]. A third group, the nano-phytoplankton, was later added to the population model and shown to dominate at intermediate chloro-

phyll concentrations [5,11,39]. Here, we parameterize a three-component phytoplankton model according to Brewin et al. [5] (Table 2; parameters below first optical depth, $\tau < 0.6$). Noise is added to the dataset by transforming the chlorophyll concentrations associated with the three size classes ($c_i$) to a random variable ($C_i$) with a log-normal distribution: $log(C_i) = \mathcal{N}(log(c_i), 1)$. The chlorophyll concentration $c_i$ is varied between 0.01–30 mg m$^{-3}$.

The absorption of CDOM ($a_{cdom}$) and detrital matter ($a_{dm}$) are modelled as an exponential function:

$$a_{cdom, \, dm}(\lambda) = a(\lambda_0) \, exp\Big[ - s(\lambda - \lambda_0)\Big] \tag{15}$$

with reference wavelength $\lambda_0$ at 440 nm and a spectral slope $s$ was randomly drawn from a normal distribution $\mathcal{N}(0.0176, 0.002)$ for CDOM and $\mathcal{N}(0.0123, 0.0013)$ for detrital matter (Figure A1) [26]. Detrital backscatter is described by a power-law according to:

$$b_{b \, dm}(\lambda) = \tilde{b}_{dm} \, b_{dm}(\lambda_0)\Big(\frac{\lambda}{\lambda_0}\Big)^{\gamma_2} \tag{16}$$

For the phase function, indicated by $\tilde{b}_{dm}$, we assume a 1.4% FF phase function and reference wavelength, $\lambda_0$, at 550 nm. The spectral slope coefficient, $\gamma_2$, randomly varies between $-2.2$ and 0.2 as a function of the chlorophyll concentration [27]. Higher spectral slopes are indicative of oligotrophic waters whereas lower slope coefficients are usually found in more eutrophic areas [40]. Variability in mass specific detrital backscatter is shown in Figure A2. Hereafter we adopt IOCCG [27] to model the co-variation between chlorophyll, CDOM and detrital matter. The IOPs are forward modelled to obtain $R_{rs}$ at 5 nm intervals using our polynomial approximation (Equations (2) and (4)). The hyperspectral dataset yields a total of 430 simulated $R_{rs}$ spectra that are used for the inversion. Retrievals with a relative error (Equation (13)) higher than 200% are discarded. All accuracy metrics for the successful retrievals are calculated on $log_{10}$ transformed variables and mean absolute error ($MAE$) and bias are backtransformed out of $log_{10}$ space according to Seegers et al. [41].

*2.6. Ocean-Color Instruments*

To demonstrate the flexibility of HYDROPTs waveband settings, we have conducted an inter-comparison of chlorophyll-a retrievals between three important ocean-color instruments: SeaWiFS, OLCI and PACE. The number of wavebands for these instruments in the 400–710 nm range are 6, 11 and 63 respectively. For the PACE sensor we have assumed equally spaced wavebands at a 5 nm resolution. Together these instruments cover the global observations of the oceans since 1997 [1] and for years to come. Of interest is the variation in retrieval accuracy that can potentially arise from the differences in band setting between these instruments. However, this exercise is not meant to be a complete analysis of the differences, since this would involve a perfect error budget of each band for each instrument ($\sigma_i$ in Equation (6)). Even with complete knowledge of instrument measurement errors, the accuracy of atmospheric correction is instrument-dependent, introducing errors and bias in the derived $R_{rs}$ [42].

The retrieval accuracy between sensors for total chlorophyll-a are evaluated using 5 metrics: slope of the linear model, bias, $MAE$, $R^2$, and fraction of successful retrievals (f-score; see Table A2). The slope, bias and MAE statistics are projected on a 0-1 scale for ease of comparison according to:

$$score = \max(\{0, 1 - |x - 1|\}) \tag{17}$$

with $x$ being the metric to be transformed. A value of one indicates a perfect score whereas a score of zero reflects a deviation of $\geq 100\%$ from a perfect score.

## 3. Results

### 3.1. Forward Model Validation

The speed and accuracy of HYDROPT are achieved by the adoption of polynomial approximations to the exact solutions of HydroLight RT simulations. In this section, we first assess and validate the accuracy of these approximations.

By varying the degree of the polynomial in Equation (2) a measure of accuracy is obtained as a function of the complexity of the model (Figure 3). The *RMSRE* accuracy metric is calculated over the $R_{rs}$ values of all 63 wavebands with the sun at a zenith angle of 30°. Increasing the model complexity reduces the prediction error in both the validation and training set up till the 4th degree. Further increasing the polynomial degree increases the prediction error and the variance, indicating model over-fitting. A 4th degree polynomial is chosen as the most parsimonious model (Figure 3). The expected average error across all wavebands is ≈ 1%.

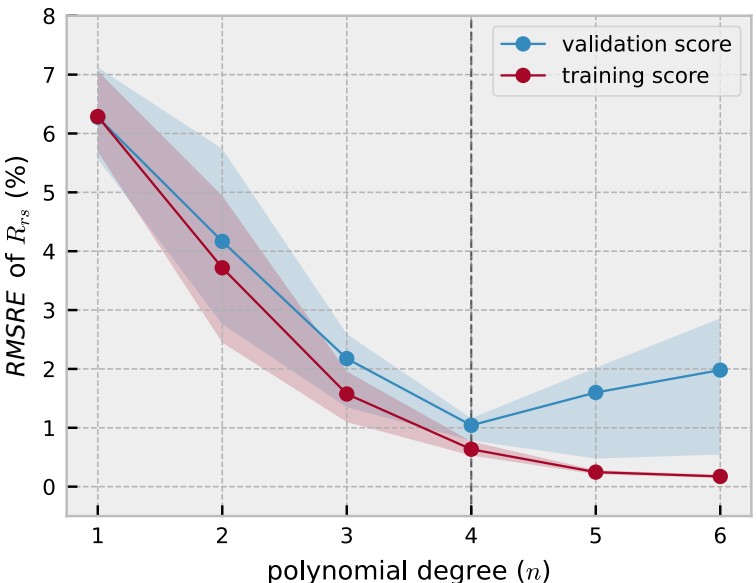

**Figure 3.** Cross-validation of the model in Equation (2) for different polynomial degrees (*n*). The model is fitted on $R_{rs}$ data at nadir with the sun at zenith angle of 30°. The accuracy metric used is the root mean squared relative error (*RMSRE*). The blue line and purple line show the validation and training score respectively. The 68% confidence envelope is shown. The dashed line indicates the most parsimonious model chosen in this study.

The average error is further decomposed to visualize the variability across wavebands. Instead of relying on the *RMSRE*, the relative error is calculated ($[\hat{R}_{rs} - R_{rs}]/R_{rs}$; with $\hat{R}_{rs}$ the reflectance predicted by HYDROPT and $R_{rs}$ the simulated reflectance calculated by HydroLight). The distribution of the relative error for 8 out of a total of 63 wavebands is visualized in Figure 4. The errors are calculated for a nadir viewing direction and solar zenith angle of 30°. The relative error per waveband shows a slight variation; however, most of the errors are well below 1% consistent with Van Der Woerd and Pasterkamp [19].

The polynomial model is validated for all possible viewing geometries. Figure 5 shows the mean relative error for all HydroLight quads ($10 \cdot 24 = 240$) at a nominal wavelength of 440 nm. The errors at nadir (see also Figure 4) are consistent with other sensor viewing angles (Figure 5) with only marginal deviations. Mean relative errors vary between 0.4% and 0.6% with the lowest values found in a sun-facing direction ($\phi = 0°$), with a slight increase to 0.6% when the sensor is directed away from the sun ($\phi = 180°$).

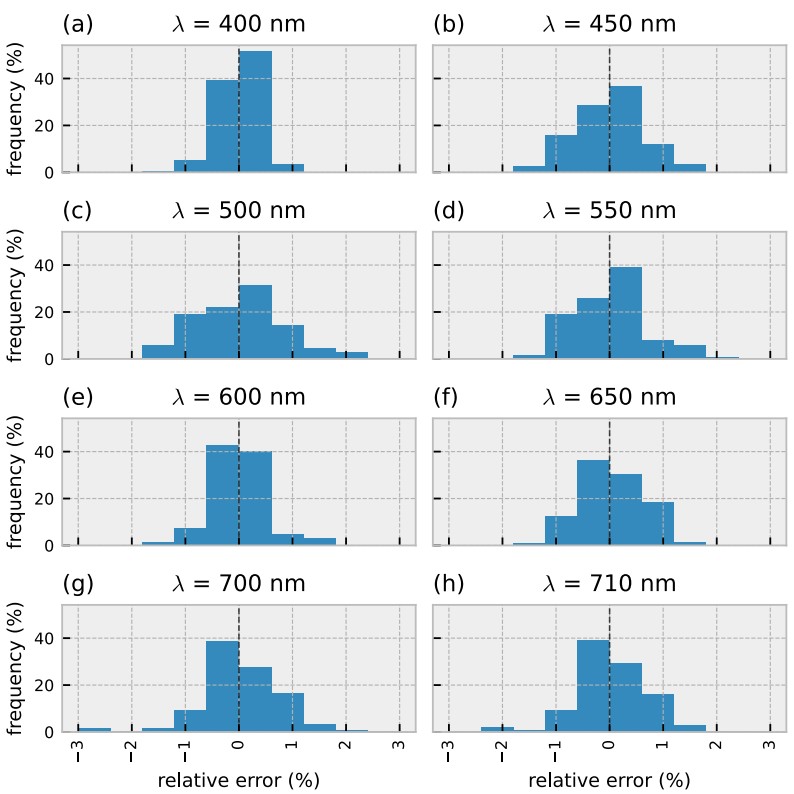

**Figure 4.** Distribution of the relative error for 8 (out of 63) wavebands in $R_{rs}$ between the HydroLight simulations and the predicted values by the 4th degree polynomial model. Validation results are for a nadir viewing angle and solar zenith angle of 30°.

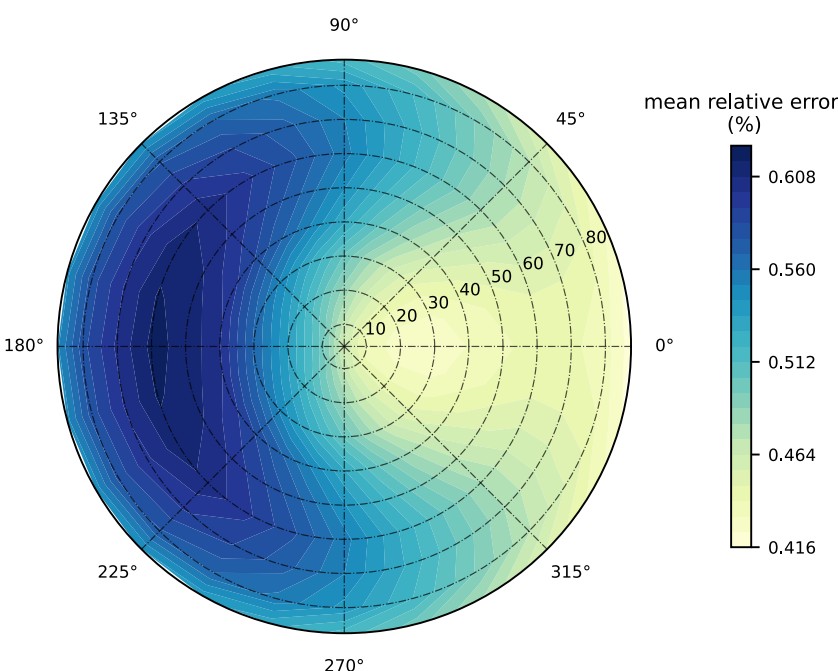

**Figure 5.** Mean relative prediction error in percent (%) in $R_{rs}$ at 440 nm for the 4th degree polynomial model for different viewing geometries. Nadir angle ($\theta$) and azimuthal angle ($\phi$) follow the default HydroLight quad layout with 10° resolution in $\theta$ and 15° resolution in $\phi$. The sun is positioned at an azimuthal angle of 0° and zenith angle ($\theta_s$) of 30°.

To test the generazibility of our forward model, we tested the predictions of HYDROPT against the independent IOCCG dataset that contains RT simulations with different phase functions for both NAP and phytoplankton. The total absorption ($a$) and backscatter ($b_b$) coefficients of the IOCCG datasets are used as inputs to the polynomial model to predict $R_{rs}$. Good agreement is reached between $R_{rs}$ from the IOCCG dataset and the predicted values from the polynomial model (Figure 6). The *MAE* indicates a relative average error of 1–2% for the three OLCI bands compared to the IOCCG $R_{rs}$ (Figure 6a–c). The maximum deviation in $R_{rs}$ are 4% for the 442.5 and 560 nm band, and 6% for the 708.75 nm band (Figure 6d–f). Together with a negligible bias and perfect $R^2$ score these results indicate robust predictions by the HYDROPT forward model.

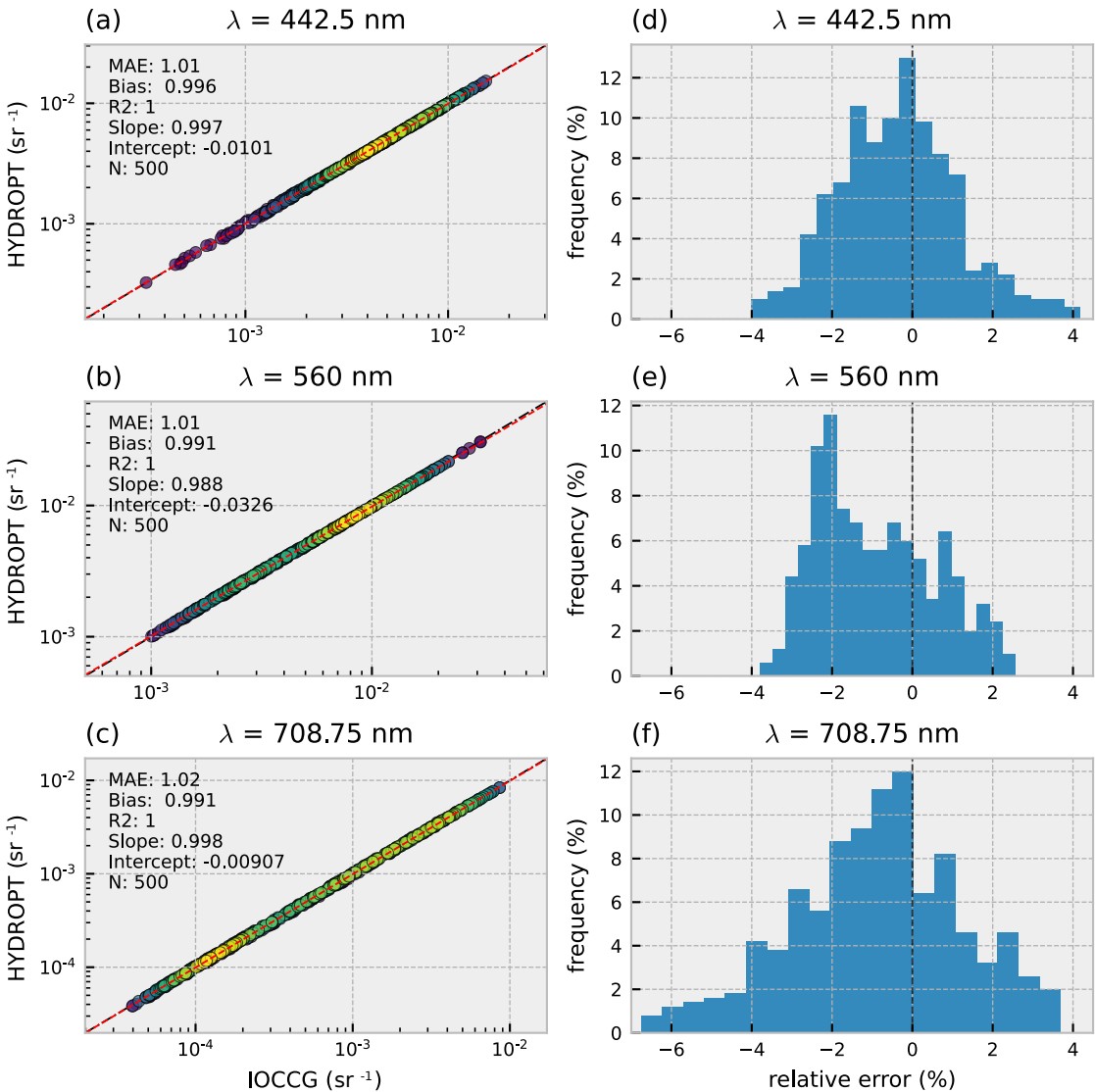

**Figure 6.** Validation of $R_{rs}$ predicted by the polynomial model vs. the IOCCG HydroLight simulations. (**a**–**c**) validation results, (**d**–**f**) distribution of the relative error (%). Results are for a nadir viewing angle with solar zenith angle ($\theta_s$) of 60°. For statistics refer to Table A2. Number of samples indicated by *N*. Black dashed line is 1:1 line, red dashed line is the linear model. Data point density is indicated by color (yellow = high, blue = low).

### 3.2. Hyperspectral Inversion

HYDROPT can use the full spectral information for the inversion of $R_{rs}$ to IOPs and concentrations of optical constituents. The inversion of $R_{rs}$ to IOPs and the comparison with the forward modelled spectra are shown in Figure 7. HYDROPT aims to minimize the difference between the forward- and inverse-modelled $R_{rs}$, which results in a near perfect

fit for the $R_{rs}$ example spectrum in Figure 7a. Since HYDROPT models $R_{rs}$ as a function of the total spectral absorption and backscatter (Equations (1) and (2)–given the viewing geometry and solar angle) it is not surprising that the total inverse-modelled absorption and backscatter are in close agreement with the forward modelled spectra (Figure 7b,c). Figure 7d–f illustrates how HYDROPT further decomposes the total absorption into the absorption by pico-, nano- and micro-phytoplankton respectively. Of the total absorption budget (at 440 nm), roughly 50% is accounted for by phytoplankton. Pico-phytoplankton dominate the phytoplankton absorption, followed by nano- and micro-phytoplankton. The quality of the match between forward- and inverse-modelled spectra follows the same order: the retrieved pico-phytoplankton absorption shows the best agreement to the forward modelled spectrum followed by nano- and micro-phytoplankton (Figure 7d–f).

Similarly, the spectral backscatter of the different optical components can be disentangled from the total backscatter budget. As such, HYDROPT can be used as a diagnostic tool to investigate the contribution of the individual optical components to $R_{rs}$.

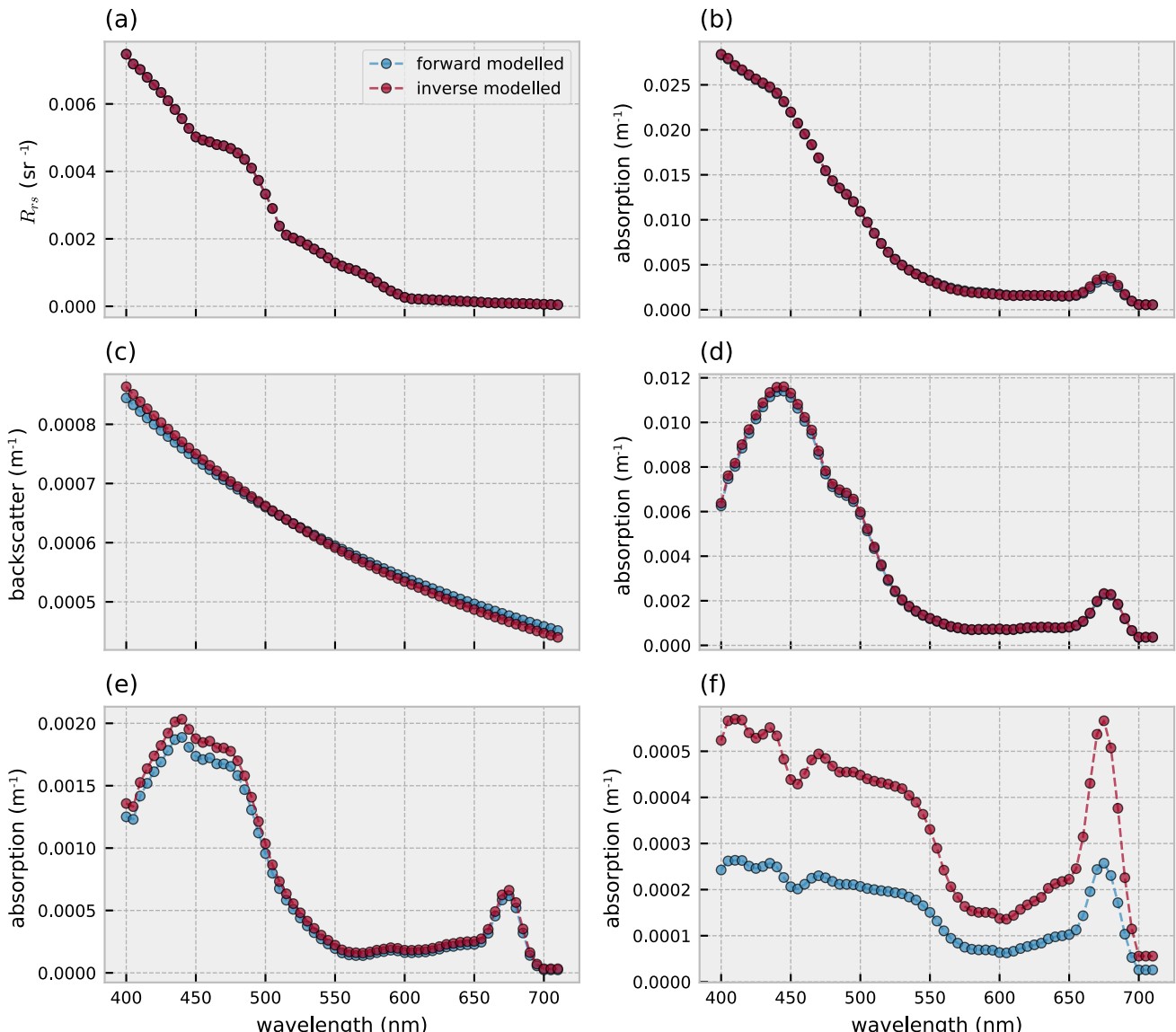

**Figure 7.** Comparison between forward- and inverse-modelled hyperspectral data for (**a**) $R_{rs}$ (**b**) total absorption excluding water (**c**) total backscatter excluding water (**d**) absorption by pico-phytoplankton (**e**) absorption by nano-phytoplankton (**f**) absorption by micro-phytoplankton. Red lines/dots indicate results of the inversion and blue lines/dots represent the forward modelled values.

### 3.3. Retrieval of Phytoplankton Size Classes

In the previous section one hyperspectral $R_{rs}$ spectrum was inverted to illustrate the decomposition into the spectral absorption by three phytoplankton size classes. Here we show the retrieval results for all 430 simulated $R_{rs}$ spectra that cover a diverse range of optical conditions and phytoplankton community composition. Pico-phytoplankton concentrations ranged from approximately $4 \cdot 10^{-3}$ to 2 mg m$^{-3}$ (Figure 8a). Of the 430 simulated spectra, HYDROPT was able to successfully invert 78% of the spectra to chlorophyll-a concentrations contained within the pico-phytoplankton size class. The average relative error expected for pico-phytoplankton is 120% (*MAE* = 2.2), taken over the entire concentration range. These deviations from ground truth values are also reflected in the negative $R^2$ and bias of 1.81, indicating that on average HYDROPT overestimates the concentration of pico-phytoplankton by roughly 80%.

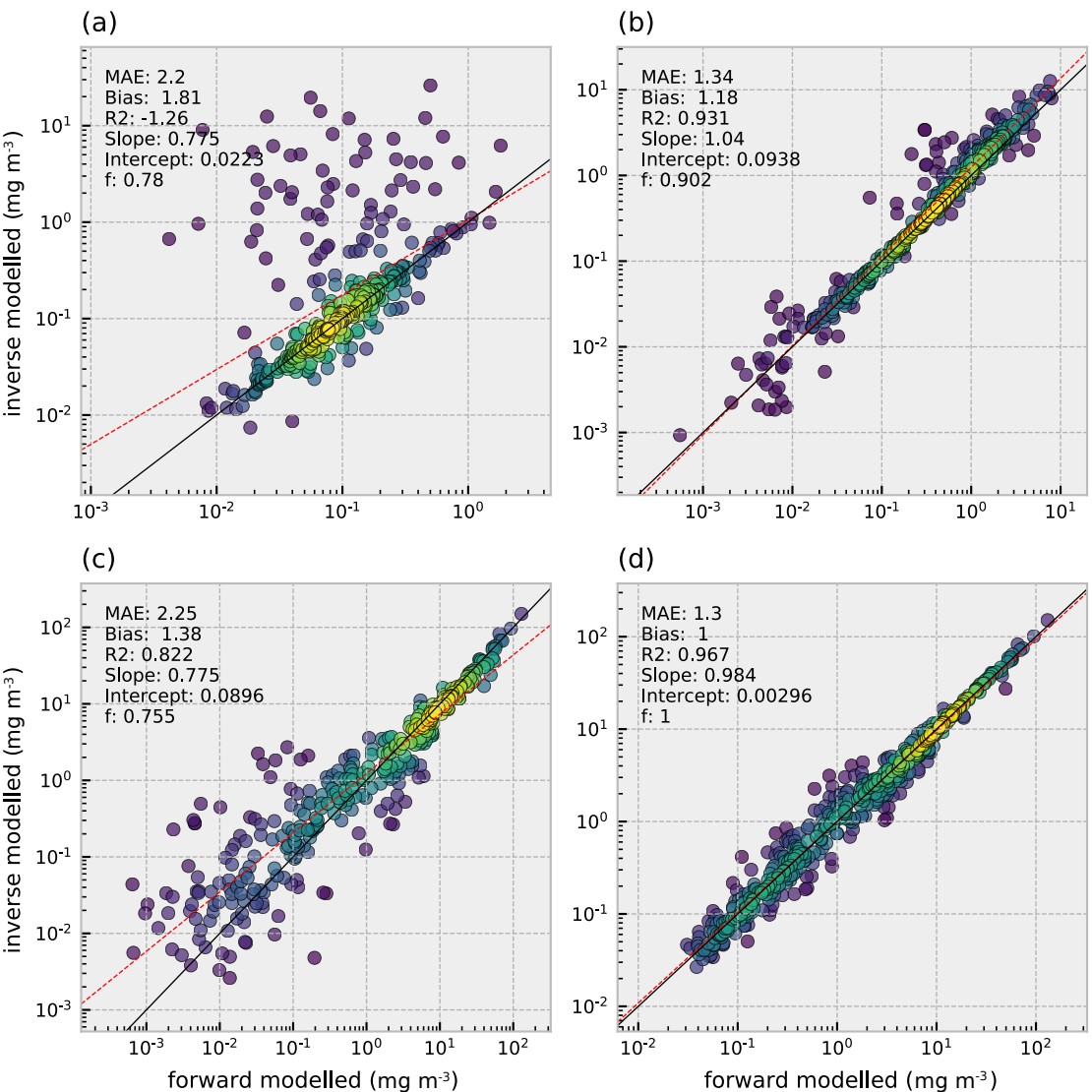

**Figure 8.** HYDROPT retrieval of chlorophyll-a concentrations for (**a**) pico-, (**b**) nano- and (**c**) micro-phytoplankton and the sum of the three size classes (**d**). For statistics refer to Table A2. The f-score indicates the fraction of reflectance spectra that could be successfully inverted. Data point density is indicated by color (yellow = high, blue = low).

The concentration in nano-phytoplankton varied between $4 \cdot 10^{-4}$ and 8 mg m$^{-3}$. Retrieval results for nano-phytoplankton were consistent with the observed concentrations (Figure 8b). HYDROPT was able to retrieve concentrations for 90% of the spectra with

an $R^2$ statistic of 0.93 and the *MAE* indicating an expected relative error of 34% over this concentration range. An average overestimation is observed of 18%.

Micro-phytoplankton retrievals were highly depended on the concentration (Figure 8c). At high chlorophyll concentrations (i.e., >10 mg m$^{-3}$) , retrievals reflected the ground truth observations more closely. However, with decreasing concentration the accuracy of the retrievals deteriorated leading HYDROPT to overestimate micro-phytoplankton more consistently (by 38%, bias = 1.38). Close to 76% of the spectra could be inverted to obtain estimates of the concentration of micro-phytoplankton with an $R^2$ of 0.82 and *MAE* of 125%.

The retrievals for total chlorophyll concentration, the sum of chlorophyll contained in the three size classes, are shown in Figure 8d. The general trend in total chlorophyll is captured well by HYDROPT with an $R^2$ of 0.97 and *MAE* of 30%. As with the retrievals for micro-phytoplankton, estimates of total chlorophyll are closer to the 1:1 line at concentrations above 10 mg m$^{-3}$. HYDROPT shows no bias in the retrievals of total chlorophyll-a (bias = 1).

The absorption and backscatter coefficients for detrital matter and CDOM could be successfully retrieved for most of the spectra (f-score 1 and 0.993 respectively) (Figure 9). Retrieval of the absorption coefficients for detrital matter showed a relatively large spread around the 1:1 line, which is reflected in an $R^2$ value of 0.67 and a *MAE* of 121%. Detrital backscatter and CDOM absorption both yielded a $R^2$ of 0.98. The *MAE* for detrital matter and CDOM were 13% and 23% respectively. Detrital backscatter and CDOM absorption retrievals were in good agreement with the observed values - only at higher values a slight but consistent overestimation is observed (bias of 6% and 14% respectively).

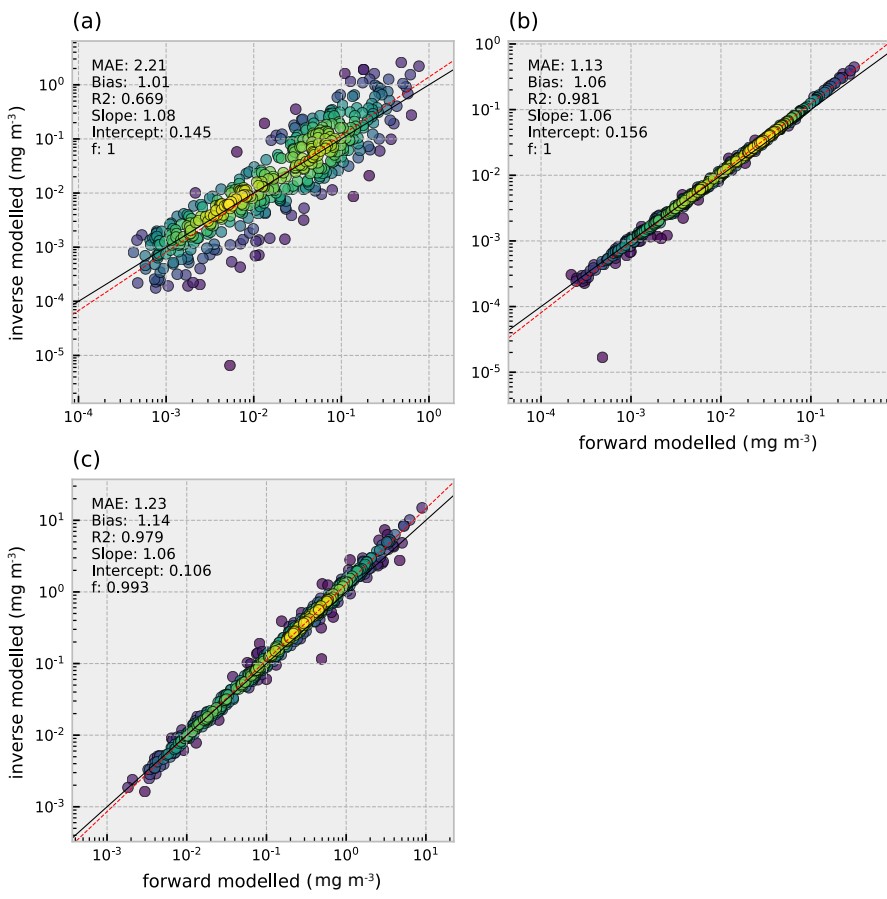

**Figure 9.** IOP retrievals for detrital matter and CDOM. (**a**) Detrital matter absorption at 440 nm and (**b**) backscatter at 550 nm. (**c**) CDOM absorption at 440 nm. For statistics refer to Table A2. The f-score indicates the fraction of reflectance spectra that could be successfully inverted. Data point density is indicated by color (yellow = high, blue = low).

### 3.4. Comparison of Multi- and Hyperspectral Retrievals

The HYDROPT framework allows for the comparison of retrievals using different band settings. Here we assess the performance of the inversion using band settings of three different sensors to retrieve the total chlorophyll-a concentration contained in pico-, nano, and micro-phytoplankton (Figure 10). The multispectral SeaWifs and OLCI and hyperspectral PACE sensor are compared to emphasize how band placement and number of spectral bands potentially influence the capability of these sensors to retrieve phytoplankton size class information. It should be kept in mind that differences in instrument measurement errors and atmospheric correction are not evaluated.

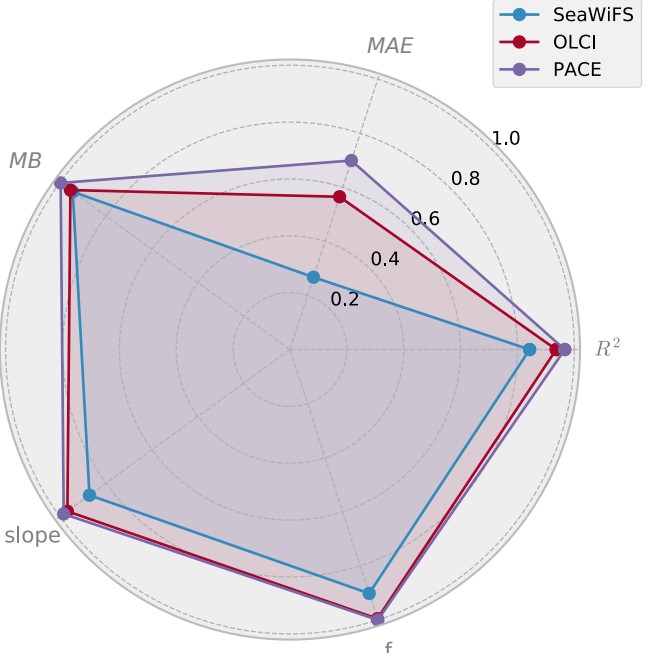

**Figure 10.** Inter-comparison of retrieval statistics for total chlorophyll-a (sum chlorophyll contained in pico-, nano- and micro-phytoplankton) between sensors. Mean bias (*MB*), mean absolute error (*MAE*) and slope are projected on a [0–1] interval according to Equation (17). A transformed statistic of 1 indicates a perfect score whereas a score of 0 indicates $\geq 100\%$ deviation from a perfect score. For calculation of statistics see Table A2. Three sensors are compared: the hyperspectral PACE sensor and multispectral OLCI and SeaWiFs sensors.

The hyperspectral PACE instrument outperformed the two other sensors on all the 5 accuracy statistics. Out of the 5 accuracy statistics, *MAE* shows the largest variation among the sensors (Figure 10). For the PACE sensor we observe a reduction in *MAE* performance of roughly 30% (i.e., an average relative error of 30% compared to ground truth), followed by 44% for OLCI and 73% for SeaWiFS. Although PACE will operate with a vastly greater number of wavebands ($k \approx 63$), the accuracy for the other 4 metrics ($R^2$, f-score, slope and mean bias) closely resembles that for the retrievals with band settings adopted by OLCI ($k = 11$). This indicates that with a limited number of bands, OLCI (and heritage sensors) achieves optimal waveband placement to retrieve total chlorophyll-a from $R_{rs}$ for water with different phytoplankton community structure. Even with 6 bands placed at SeaWifs waveband center it is possible to retrieve size class information, albeit with further reduction in accuracy and in absence of errors in sensor measurements and atmospheric correction.

## 4. Discussion

Previous attempts have been made to derive phytoplankton community structure from optical measurements, but these approaches either relied on empirical models [2,5,43–46] or the decomposition of (remotely estimated) $a_{ph}$ or $b_{bp}$ [9,13,14,39,47,48]. The advantage of empirical models are speed and ease of use [49]. In addition, empirical models, such as abundance-based models, also only require remotely retrieved chlorophyll estimates to derive functional types [43] and/or size classes [5,46]. Other empirical models directly derive community structure from $R_{rs}$ [44,45]. However, empirical models require reparameterization for different biogeographic regions [5,50–52] and are biased toward the training data used to build the model.

Retrieval models that adopt absorption and scattering-based approaches usually require less empiricism than the previously discussed models and adopt radiative transfer principles [3,6]. However, most methods use only part of the spectral information for the inversion and only exploit knowledge of the relationship between cell size and either the absorption or backscatter coefficients. The inter-dependency between $R_{rs}$ and the absorption and backscatter budget is not fully captured.

One of the few attempts to directly retrieve community composition from $R_{rs}$ using radiative transfer principles is Werdell et al. [15]. Werdell et al. [15] used the Gordon approximation for $R_{rs}$ for waters with phytoplankton community consisting of two different species. The IOPs are forward modelled and the best fit to the measured $R_{rs}$ is determined through nonlinear optimization (Levenberg-Marquardt). In this case, full spectral information is used for the inversion as well as knowledge of both the absorption and backscatter characteristics of the optical components. However, using their method, Werdell et al. [15] note that further decomposing the contribution of each phytoplankton group to the total IOP budget significantly degrades the accuracy of the retrievals. Their effort serves as an example of the difficulty of directly retrieving multiple phytoplankton groups from $R_{rs}$, without relying on empirical methods.

Here we applied the updated HYDROPT inversion framework to $R_{rs}$ spectra and showed that under ideal circumstances it is possible to directly retrieve the concentration of multiple phytoplankton groups in addition to CDOM and detrital matter (Figures 8 and 9). Possible explanations for the discrepancy between our results and those of Werdell et al. [15] are (1) chlorophyll-a specific IOPs used by Werdell et al. [15] are too similar to estimate the separate contribution of the two phytoplankton groups to the total IOP budget, and/or (2) in contrast to Werdell et al. [15], we incorporated different backscatter spectra for the phytoplankton groups (Figure 2). So detailed knowledge of both the absorption and backscatter spectra could potentially aid in improved retrievals of different phytoplankton groups from $R_{rs}$.

To test the feasibility of HYDROPT to invert multi- and hyperspectral reflectance measurements to obtain phytoplankton size structure information we created a dataset with simulated $R_{rs}$ spectra. In this research we have shown that it is possible to obtain concentration estimates of the three phytoplankton size classes from reflectance spectra under ideal conditions (e.g., no sensor noise, perfect atmospheric correction). By supplying detailed absorption and backscatter spectra of 3 phytoplankton size classes, CDOM and detrital matter (Figures 1, 2, A1 and A2), HYDROPT can optimize the spectral fit and estimate the contribution of the optical constituents to $R_{rs}$. HYDROPT additionally calculates error statistics for the retrievals. A retrieval was considered valid when the relative error is within 200%. Nano-phytoplankton could be retrieved for 90% of the spectra, followed by pico-phytoplankton (78%) and micro-phytoplankton (76%) (Figure 8). Ambiguity in retrievals could be due to several factors, including chlorophyll biomass [17] but also the absorption and scattering properties [34]. However, future research on the sensitivity of $R_{rs}$ to changes in phytoplankton community composition should elucidate the limits on the retrieval accuracy using semi-analytical approaches such as HYDROPT.

Many empirical and semi- analytical retrieval algorithms exist [3,6]. However, most algorithms are configured for specific wavebands making it difficult to apply these algo-

rithms to different sensors to derive the same bio-optical parameters–hence obstructing data fusion and inter-comparison experiments [53,54]. The retrieval inter-comparison experiment (Figure 10) shows how HYDROPT can easily be applied to different ocean-color sensors to invert $R_{rs}$ to the IOPs and concentration of optical components of interest. The retrieval performance of OLCI, and to a lesser extent SeaWiFS show that with a limited number of wavebands and correct band placement total chlorophyll-a and phytoplankton size structure can be obtained from remotely measured $R_{rs}$. However, given the number of wavebands and thus the available degrees of freedom for the inversion, hyperspectral measurements as would be obtained from PACE allow many more parameters to be estimated from $R_{rs}$ (e.g., larger set of phytoplankton groups, aerosols, atmospheric composition etc. [22]) or target distinct optical characteristics such as absorption features of accessory pigments to distinguish different phytoplankton taxa [4,7].

For the accurate inversion of $R_{rs}$ spectra, HYDROPT relies on (1) a forward model rooted in radiative transfer theory, (2) specification of IOP spectral models for the region of interest, and (3) fast optimization routines that find the best fit to the measured $R_{rs}$ spectrum. We shortly discuss these three characteristic features of the HYDROPT framework below.

In most reflectance models the bidirectional nature of $R_{rs}$ is accounted for by the so-called $f/Q$ factor [23,55]. Interestingly, the $f/Q$ factor exhibits spectral dependence due to the relative importance of molecular scattering over particle scattering and therefore changes in the total VSF with wavelength [24]. The dependency of the $f/Q$ factor on wavelength is complex and is further influenced by the effects of CDOM absorption and number of scattering events in the water column [56,57]. HYDROPT accounts for changes in the bidirectional effects with wavelength by fitting the forward model on HydroLight simulations at every 5 nm in the visible domain (400–710 nm). Furthermore, variability in $R_{rs}$ due to sensor- and sun position [24,25,58] is resolved by refitting the polynomial model (Equation (2)) for different sensor viewing geometries and solar zenith angles.

Our validation benchmark shows that HYDROPT accurately predicts $R_{rs}$ with an average error of 1% compared to the RT simulations (Figure 3), consistent with Van Der Woerd and Pasterkamp [19]. The relative errors in $R_{rs}$ show a weak dependency on sensor position, ranging between 0.4–0.6% for the 440 nm band (Figure 5). The importance of the VSF in predicting $R_{rs}$ has been noted in several studies, especially for phytoplankton blooms and highly scattering environments [33,56,59]. Validation against the independent IOCCG dataset shows that HYDROPT is capable of accurately predicting $R_{rs}$ with a maximum deviation of 4–6%, even for waters that assume a different VSF (Figure 6).

The validation of HYDROPT shows that the forward calculations can be applied to waters for a wide range of optical conditions. Inverting $R_{rs}$ for these waters is achieved by finding the best spectral fit through nonlinear optimization routines in conjunction with our forward model. The updated HYDROPT framework allows the configuration of multiple optical components and various optimization routines to invert $R_{rs}$ such as Levenberg-Marquardt (LM) [15,19,60], Nelder–Mead [17] and simulated annealing [61]. In addition, when signal to noise (SNR) levels are well characterized for the spectral bands, the importance of the individual bands during the optimization can be accounted for by assigning appropriate weights to every waveband ($\sigma_i$ in Equation (6)) [19].

HYDROPT was originally developed with the aim to retrieve chlorophyll-a concentrations from optically complex waters where absorption and scattering by CDOM and sediment dominate [19]. We like to stress that HYDROPT is now also able to give a more in-depth analysis of complex coastal waters by including multiple spectral models for CDOM and sediments originating from different sources. Given the natural variability in absorption and backscatter characteristics for phytoplankton, CDOM and particulate matter, potential future improvements could be the addition of Bayesian optimization routines to the HYDROPT framework [62]. Instead of relying on average spectral IOP models for the inversion (red lines in Figures 1, 2, A1 and A2) a probabilistic approach would allow to account for variability in the mass specific IOP reference spectra (blue lines).

The $R_{rs}$ predicted by HYDROPT showed a higher relative error for the IOCCG data (Figure 6) than for our own simulated spectra using HydroLight (Figure 4). Furthermore, the prediction errors of the IOCCG data are more pronounced at longer wavelengths, reaching up to +6% at 708 nm (Figure 6). These results indicate that accounting for changes in the VSF is an important step toward improvement of highly accurate $R_{rs}$ predictions. Park and Ruddick [58] included an extra parameter in their forward model to account for changes in the VSF when transitioning between case-I and case-II waters. In the future, HYDROPT could be adopted in a similar way. The inclusion of shape details of the VSF into the forward model [63,64] would be a significant improvement and allow HYDROPT to accurately predict $R_{rs}$ for the large optical diversity encountered in the world's oceans, coastal waters, rivers and lakes.

## 5. Conclusions

In this paper, we demonstrated the ability of the HYDROPT framework to retrieve 3 phytoplankton size classes from simulated reflectance spectra using the principles of radiative transfer theory (Figure 8). Moreover, HYDROPT can decompose the contribution of all optically active components to $R_{rs}$ including CDOM and detrital matter (Figure 9). Our framework exhibits fast and robust prediction of $R_{rs}$ with an expected average error of $\approx$1% compared to HydroLight's radiative transfer solutions (Figure 3) while accounting for bidirectional effects across wavebands (Figure 4) and different solar-viewing geometries (Figure 5).

The flexible configuration of wavebands makes it possible to apply the framework to past, current and future multi- and hyperspectral missions such as PACE (Figure 10). In addition to satellite measurements, HYDROPT is also able to process in-situ above-water $R_{rs}$. In our simulated hyperspectral $R_{rs}$ dataset we were not able to account for all the variability that can be encountered in the real world such as errors in the measurements, fluorescence, Raman scattering and imperfect atmospheric correction to name a few. We have made HYDROPT available as an open-source Python package [21] and invite the aquatic remote sensing community to apply the framework to existing [65,66] and future multi- and hyperspectral measurements [22] and benchmark the performance under various real-world scenarios.

**Author Contributions:** Conceptualization, H.J.V.D.W. and T.H.; methodology, H.J.V.D.W. and T.H.; software, T.H.; validation, T.H.; data curation, T.H.; writing—original draft preparation, T.H. and H.J.V.D.W.; writing—review and editing, T.H. and H.J.V.D.W.; visualization, T.H. All authors have read and agreed to the published version of the manuscript.

**Funding:** This research was funded by the Dutch Research Council (NWO) under grant no. ALW-GO 14-06.

**Institutional Review Board Statement:** Not applicable.

**Informed Consent Statement:** Not applicable.

**Data Availability Statement:** The HYDROPT framework is openly available as a Python package on Github at https://github.com/tadz-io/hydropt under DOI number: 10.5281/zenodo.4782707. HYDROPT version 0.2.3 was used in this study. The synthetic hyperspectral dataset presented in this study (v1.0.0) is available on Zenodo at 10.5281/zenodo.4783289.

**Acknowledgments:** We thank Robert Frouin and Jing Tan for hosting Tadzio Holtrop at Scripps Institution of Oceanography (San Diego, CA, USA) during the development phase of the HYDROPT model and for providing feedback and inspiration for future research. We also thank Philipp Grötsch (Gybe, USA) for insightful questions and discussions related to the HYDROPT model and Jeroen Aerts and Jef Huisman for their guidance in the early stages of this work and valuable suggestions to improve this manuscript. Hans van der Woerd also sincerely likes to thank Jeroen Aerts for the opportunity to return to the Institute for Environmental Studies in 2016.

**Conflicts of Interest:** The authors declare no conflict of interest. The funder had no role in the design of the study; in the collection, analyses, or interpretation of data; in the writing of the manuscript, or in the decision to publish the results.

## Abbreviations

The following abbreviations are used in this manuscript:

| | |
|---|---|
| RT | Radiative transfer |
| HYDROPT | HydroLight Optimization |
| IOP | Inherent optical properties |
| CDOM | Colored dissolved organic matter |
| VSF | Volume scattering function |
| FF | Fournier–Forand phase function |
| PACE | Plankton, Aerosol, Cloud, ocean Ecosystem |
| MERIS | MEdium Resolution Imaging Spectrometer |
| CZCS | Coastal Zone Color Scanner |
| OLCI | Ocean Land Color Instrument |
| SeaWiFS | Sea-viewing Wide Field-of-view Sensor |
| $R_{rs}$ | Observed remote sensing reflectance-either from HydroLight simulations or synthetic dataset |
| $\hat{R_{rs}}$ | Predicted remote sensing reflectance by the HYDROPT forward model |
| $R'_{rs}$ | Natural log-transformed remote sensing reflectance |
| $a$ | Absorption coefficient |
| $b_b$ | Backscatter coefficient |
| $\tilde{b}$ | Backscatter ratio |
| $a^*, b_b^*$ | chlorophyll-a- or mass specific absorption and backscatter coefficient |
| $\rho_i$ | Concentration of constituent $i$ or absorption by CDOM at reference wavelength |
| $\hat{\rho}_i$ | Estimated concentration or absorption of optical constituent $i$ |
| RMSRE | Root mean squared relative error |
| MAE | Mean absolute error |

## Appendix A

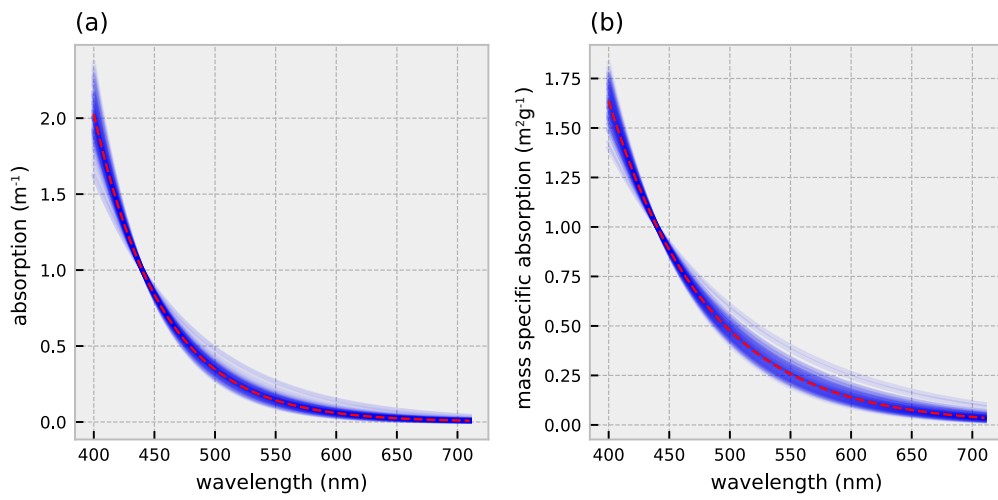

**Figure A1.** Spectral absorption for CDOM and detrital matter. (**a**) CDOM absorption. (**b**) mass specific detrital absorption. CDOM and detrital matter absorption are shown for a range of realistic spectral slopes (blue lines) [26]. Red dashed lines indicate averaged spectral absorption used for the inversion.

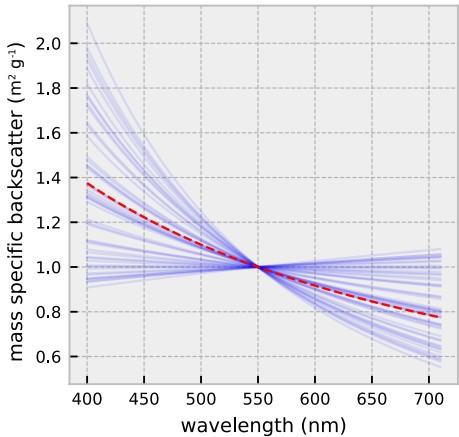

**Figure A2.** Mass specific detrital backscatter. Spectral slopes varies randomly between $-0.2$ and 2.2 [27]. Red dashed line indicates spectral backscatter with averaged spectral slope ($\gamma_2 = 1$) used for the inversion.

**Table A1.** HydroLight model parameters.

| Parameter | Value | Units | Notes | References |
|---|---|---|---|---|
| | | | Case-II bio-optical model | |
| **Sea-water** | | | | |
| Absorption | - | m$^{-1}$ | See references | Pope and Fry [67] for >550 nm Mason et al. [20] <550 nm |
| Phase function | - | sr$^{-1}$ | See reference | Equation 3.30 in Mobley [68] |
| Elastic scattering | - | m$^{-1}$ | See reference | Equation. 3.31 in Mobley [68] |
| Inelastic (Raman) scattering | - | m$^{-1}$ | No inelastic scattering | |
| **Phytoplankton** | | | | |
| Absorption | - | m$^{-1}$ | See reference for spectral absorption | Prieur and Sathyendranath [69] |
| Phase function | - | sr$^{-1}$ | Fournier-Forand (1.4% backscatter ratio) | |
| Scattering | - | m$^{-1}$ | Spectral backscatter according to: $bb_{phyto} = 0.00255 * [Chl]^{0.471}$ | |
| Fluorescence | - | - | No chlorophyll fluorescence | |
| Concentration | 0.01–31.62 | mg m$^{-3}$ | | |
| **Colored dissolved organic matter** | | | | |
| $a_{CDOM}(440)$ | 0.005–1 | m$^{-1}$ | | |
| Absorption | - | m$^{-1}$ | See reference for spectral absorption | Babin et al. [26] |
| Slope | 0.017 | nm$^{-1}$ | Exponential decay function with reference at 440 nm | Babin et al. [26] |
| **Non-algal particles** | | | | |
| concentration | 0.01–100 | g m$^{-3}$ | | |
| Absorption | - | m$^{-1}$ | See reference for spectral absorption | Babin et al. [26] |
| Slope (spectral absorption) | 0.0123 | nm$^{-1}$ | Exponential decay function with reference at 443 nm | Babin et al. [26] |
| Phase function | - | sr$^{-1}$ | Fournier-Forand (1.4% backscatter ratio) | |
| Backscatter | - | m$^{-1}$ | See reference for spectral backscatter | Babin et al. [70] |
| Slope (spectral backscatter) | $-1$ | nm$^{-1}$ | power-law with reference wavelength at 550 nm | Babin et al. [70] |
| Sea-surface boundary model | | | | |
| Wind speed | 5 | m s$^{-1}$ | | |
| Real index of refraction of water | 1.34 | - | Wavelength indepedent | |
| Atmospheric model (RADTRAN-X) | | | | |
| Solar zenith angle | 0–80 | degrees | 10 degree intervals | |
| Cloud cover | 0 | percent | Clear sky | |
| Earth-sun distance | - | - | Yearly average | |
| 24-h averaged wind speed | 5 | m s$^{-1}$ | | |
| Horizontal visibility | 15 | km | | |
| Relative humidity | 80 | percent | | |
| Precipitable water content | 2.5 | cm | | |
| Total ozone | 300 | Dobson units | Yearly average | |
| Airmass type | 1 | - | Marine | |
| Bottom reflection model | | | | |
| Depth | - | m | Infinitely deep (no bottom reflection) | |
| Output | | | | |
| Wavebands | 400–710 | nm | 5-nm resolution | |
| Radiance (upwelling) | - | W sr$^{-1}$ m$^{-2}$ nm$^{-1}$ | Radiance distribution for all HydroLight quads | |
| Irradiance (downwelling) | - | W m$^{-2}$ nm$^{-1}$ | | |

**Table A2.** Definition of retrieval statistics. $x$ are the $log_{10}$-transformed forward modelled values, $y$ are the estimated $log_{10}$-transformed values from the inversion. Total number of successful retrievals are indicated by $n$ (with $\delta \leq 2$; see Equation (13)) and total number of $R_{rs}$ spectra in the dataset by $n^*$ (=430).

| Abbreviation | Definition | Formula |
|---|---|---|
| *MAE* | Mean absolute error | $10^{\left(\dfrac{\sum\limits_{i=1}^{n}\lvert y_i - x_i \rvert}{n}\right)}$ |
| *MB* | Mean bias | $10^{\left(\dfrac{\sum\limits_{i=1}^{n} y_i - x_i}{n}\right)}$ |
| $R^2$ | Coefficient of determination | $1 - \dfrac{\sum\limits_{i=1}^{n}\left[y_i - x_i\right]^2}{\sum\limits_{i=1}^{n}\left[x_i - \overline{x}\right]^2}$ |
| slope | slope of linear regression model | $\dfrac{\sum\limits_{i=1}^{n}\left[x_i - \overline{x}\right]\left[y_i - \overline{y}\right]}{\sum\limits_{i=1}^{n}\left[x_i - \overline{x}\right]^2}$ |
| f | fraction of successful retrievals | $n/n^*$ |

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
