# Peer review of "HYDROPT: An Open-Source Framework for Fast Inverse Modelling of Multi- and Hyperspectral Observations from Oceans, Coastal and Inland Waters"

_remotesensing, doi:10.3390/rs13153006_

Round 1
Reviewer 1 Report
The research that the authors have done is extremely important and useful. The software proposed by them could really be in demand for diagnostics of inland waters using satellite data. However, as the authors themselves note, a big problem arises here. This is an atmospheric correction. For inland waters, this problem is much more difficult than for open areas of seas and oceans, especially under conditions of intense bloom with chl-a concentrations of 200-400 mg / m^3. In other words, it is difficult to be sure of the ideality of the reflectance spectrum after atmospheric correction for its subsequent use for the recovery of water components.
Nevertheless, I note once again that the authors' idea is absolutely useful. However, despite this, the article is rather poor. There is no new physics in it, no convincing results, there are questions about applicability at high concentrations of chlorophyll-a (> 100). In my opinion, the article is not very suitable for the selected journal. However, if Editors disagree with me, then I have no comments on the article: the material is presented consistently, the English language is good enough, the discussion is sufficient.
Reviewer 2 Report
Very good! This was well presented with a great number of references, clean and understandable figures and captions. The appendix, tables explaining terms and acronyms, everything was very readable and well-laid out for someone who may not understand the topic as well as the authors.
I feel that this is a polished paper in its present form, the authors appear to have gone out of their way to lay out the definitions for every step and use concise, clear graphics and excellent labels.
This work is not in my direct field, but is adjacent enough that I feel comfortable saying I would have detected any short-cuts with the data.
Thank you for a great, informative read!